# Potential Industrial Applications and Commercialization of Microalgae in the Functional Food and Feed Industries: A Short Review

**DOI:** 10.3390/md17060312

**Published:** 2019-05-28

**Authors:** Franciele Camacho, Angela Macedo, Francisco Malcata

**Affiliations:** 1LEPABE—Laboratory of Process Engineering, Environment, Biotechnology and Energy, University of Porto, Rua Dr. Roberto Frias, s/n, 4200-465 Porto, Portugal; franciele@fe.up.pt (F.C.); fmalcata@fe.up.pt (F.M.); 2UNICES-ISMAI—University Institute of Maia, Av. Carlos Oliveira Campos, 4475-690 Maia, Portugal; 3Department of Chemical Engineering, University of Porto, Rua Dr. Roberto Frias, s/n, 4200-465 Porto, Portugal

**Keywords:** human health, animal health, nutrition, functional food, prebiotic, probiotic

## Abstract

Bioactive compounds, e.g., protein, polyunsaturated fatty acids, carotenoids, vitamins and minerals, found in commercial form of microalgal biomass (e.g., powder, flour, liquid, oil, tablet, or capsule forms) may play important roles in functional food (e.g., dairy products, desserts, pastas, oil-derivatives, or supplements) or feed (for cattle, poultry, shellfish, and fish) with favorable outcomes upon human health, including antioxidant, anti-inflammatory, antimicrobial, and antiviral effects, as well as prevention of gastric ulcers, constipation, anemia, diabetes, and hypertension. However, scale up remains a major challenge before commercial competitiveness is attained. Notwithstanding the odds, a few companies have already overcome market constraints, and are successfully selling extracts of microalgae as colorant, or supplement for food and feed industries. Strong scientific evidence of probiotic roles of microalgae in humans is still lacking, while scarce studies have concluded on probiotic activity in marine animals upon ingestion. Limitations in culture harvesting and shelf life extension have indeed constrained commercial viability. There are, however, scattered pieces of evidence that microalgae play prebiotic roles, owing to their richness in oligosaccharides—hardly fermented by other members of the intestinal microbiota, or digested throughout the gastrointestinal tract of humans/animals for that matter. However, consistent applications exist only in the dairy industry and aquaculture. Despite the underlying potential in formulation of functional food/feed, extensive research and development efforts are still required before microalgae at large become a commercial reality in food and feed formulation.

## 1. Introduction

Microalgae are ancestral living organisms that constitute the basis of aquatic food chains. They are a phylogenetically diverse group, encompassing a number of different phyla and classes of organisms, in some cases, cyanobacteria are also included [1,2]. Microalgae grow well not only in freshwater, seawater, and hypersaline environments, but also in moist soils and rocks [3].

Microalgae have to date found a number of industrial applications—examples of success include formulation of food [4], feed [5], cosmetics [6], health products [7], and fertilizers [8], as well as tools for wastewater treatment [8,9] and biofuel production [5]. However, most technological research findings have failed to reach commercial level due to constraints encompassing, namely: (i) small market size; (ii) production at uncompetitive cost, compared to alternative products obtained either via chemical synthesis, or directly as a result of metabolism by other microorganisms (e.g., fungi, bacteria), or even as extracts from fossil raw materials; and (iii) tighter regulatory constraints in terms of quality specifications, safety assurance, and minimization of environmental impact [10,11].

Microalgae may be regarded as a promising food or feed ingredient, owing to their nutritional features [12]—a trait highly dependent upon microalga own composition, and amount thereof in the diet(s). Furthermore, the said nutritional compounds depend on the species used and growth conditions provided, namely in terms of light, temperature, and nutrient profile.

A wide spectrum of biologically active compounds have been found in microalgal biomass—in the form of protein, polyunsaturated fatty acids (PUFAs), pigments, vitamins, and minerals, or as extracellular compounds, such as oligosaccharides [13,14,15]. The associated efficiency of (bio) synthesis is greater than that observed in terrestrial crop plants.

A number of studies have claimed benefits of such microalgal compounds upon human health—bearing anticancer, anti-inflammatory, antioxidant, antimicrobial, and anti-obesity capacities, further to hypocholesterolemic character. Hence, they may serve as nutraceuticals [15], and their market value in food is expected to rise on the short run [16].

Microalgae have also been increasingly tested either as nutraceutical by the feed industry—especially for aquaculture, to improve immune response of marine animals. Unfortunately, commercial use has faced hardships as per the high production costs pertaining to concentration and storage [17]. Additionally, industrial use of microalgae as prebiotics in feed formulation has faced several challenges and novel or improved processing techniques are still required to lower production costs [18].

The main goal of this short review is to highlight recent research studies and patents, encompassing (actual and proposed) industrial applications of microalgae/cyanobacteria — specifically in the functional food and feed areas, which bear on their own a major potential for improvement of human and/or animal health. The gap/bridge between research development and actual commercialization of such high-value products from microalgae is also discussed to some extent.

## 2. Microalgae as Functional Food

Food is originally intended to respond to hunger, and thus provide nutrients needed for human survival. According to United Nation figures, the global population is expected to grow by almost 50% since 2000, to 9.5 billion in 2050 [19]. Not only will the required amount of food increase, but also the type of foods sought and their relative contribution to diet(s). Worldwide demand for animal-derived protein will likely double by 2050 [20], thus raising concerns regarding (sustained) security and safety. Meanwhile, negative environmental impacts have been occurring, e.g., generation of larger and larger emissions of greenhouse-effect gases, besides the excessive amounts of freshwater and land occupation required.

Large-scale cultivation of microalgae started in the 1960s in Japan, where *Chlorella* sp. were used as a food additive. In the 1970s and 1980s, industrial production of microalgae expanded to USA, China, Taiwan, Australia, India, Israel, and Germany—with an emphasis on *Spirulina* sp., and *Chlorella* sp. in more recent years. The current combined production adds up to ca. 5000 and 2500 tons of dry biomass, respectively, and has contributed to address the so-called “protein gap” [21].

As a source of proteins, microalgae other than *Chlorella vulgaris* or *Arthrospira plantesis* (former *Spirulina*) as is the case of *Dunaliella salina***,**
*Haematococcus pluvialis,* and *Phaeodactylum tricornutum*, have been employed by the food industry. Unfortunately, utilization of microalgae as a dietary protein source is still poorly developed in Europe because it requires prior development of novel value chains—with special attention paid to such issues as production costs (algal protein *versus* e.g., soy or lupin), food safety (especially in view of strict European legislation on the subject), scalability of processes, and consumer acceptance at large [22,23].

Meanwhile, demand for functional foods has boomed in recent decades, due to consumers’ growing awareness of food impact upon human’s health. Functional foods (or nutraceutical components in food) trigger, by definition, beneficial effects upon physiological functions, improve the well-being and the health of consumers, and reduce the risk of illness [13]. Therefore, regular inclusion of functional food in the diet promotes quality of life, and will eventually reduce costs of health care for the population at large—characterized by an ever growing life expectancy [24]. The functional value of microalgae utilized as food arises from their high contents of proteins, polyunsaturated fatty acids, polysaccharides, pigments, vitamins, minerals, phenolic compounds, volatile compounds, and sterols [21].

Microalgae are relevant sources of long-chain polyunsaturated fatty acids and they have accordingly been employed in the food industry as supplements. Microalgae are indeed able to synthesize members of the omega 6 family (ω6)—which include linoleic acid, γ-linolenic acid (GLA), and arachidonic acid (ARA), as well as of the omega 3 family (ω3)—which include linolenic acid, eicosapentaenoic acid (EPA), and docosahexaenoic acid (DHA). DHA and EPA are associated with reduction of complications in cardiovascular effusions, arthritis, and hypertension; they also exhibit relevant hypolipidemic activity, for reducing triglycerides and increasing high-density lipoprotein cholesterol [25].

DHA is also relevant for development and functioning of the nervous system. ARA and EPA are responsible for aggregative and vasoconstricting action of platelets, and anti-aggregative and vasodilator effects in the endothelium—besides chemotactic action in neutrophils [25,26]. The pigments of microalgae belong, in general, to one of three classes: i) chlorophylls; ii) carotenes and xanthophylls; and iii) phycobiliproteins, such as c-phycocyanin and allophycocyanin. They are responsible for several health benefits, namely anti-inflammatory, antihypertensive, anticancer, antioxidant, antidepressing, and antiaging features [27,28,29] (see Table 1); however, their primary application is as food colorants. Human beings are incapable of synthesizing these pigments, so they rely solely upon dietary intake thereof. β-Carotene in *Arthrospira* and astaxanthin in *Haematococcus pluvialis* may account for more than 80% (relative to biomass) of the total carotenoids in cells [30]. Several studies focused on cell accumulation and product yield optimization by strains of these two green algal species—further to attempts to improve their extractability. Furthermore, astaxanthin and β-carotene have experienced a strong and ever growing market demand. According to forecasts by Zion Market Research Global in 2016 [31], the market value of carotenoids is anticipated to evolve, between 2016 and 2021, at a (compound) annual growth rate of 3.5%, thus reaching an expected revenue of USD 1.52 billion by 2021. Due to their intrinsically high levels of endocellular accumulation of carotenoids along with the projected market demand, *Dunaliella salina* and *Haematococcus pluvialis* are generally considered as promising microalgal sources for industrial bioprocessing.

β-Carotene is an intensely-colored orange pigment, abundant in green leafy plants (e.g., parsley, spinach, broccoli), certain fruits (e.g., mandarin, peach), and several vegetables (e.g., carrot, pumpkin) [66]. It is a precursor (or inactive form) of vitamin A, which is synthesized from carotenoids via the catalytic action of β-carotene 15,15′-monooxygenase 1. Vitamin A is a widely-recognized factor toward child health and survival. Its deficiency causes disturbances in vision, and leads to lung, trachea, and oral cavity pathologies. Other biological roles performed by β-carotene include absorption of light energy, transport of oxygen [67], enhancement of in vitro antibody production, and antitumor, antioxidant, and anti-inflammatory activities [68]. Among all natural sources of carotenoids, marine microorganisms have emerged as the easiest to handle. Therefore, the European Commission and other international bodies have been contracting research in blue biotechnology—concerned with exploration and exploitation of the biodiversity of marine organisms, aiming at development of novel products [69]. A significant number of research groups are in fact working, in a coordinated manner, to strengthen knowledge about carotenoids, and find novel natural sources; this the case of such networks as International Carotenoid Society [70], Eurocaroten [71], IBERCAROT [72], and CaRed [73].

Seafood (such as salmon, shrimps, crabs, lobsters, crayfish, and trout) is the major source of astaxanthin in the human diet, but their amount is usually insufficient. The U.S. Food and Drug Administration (FDA) approved astaxanthin as nutritional supplement in 1999, owing to beneficial effects stemming from its potent anti-oxidant features [74]. Among the various health benefits claimed, viz. anti-diabetic, anti-oxidative, anti-inflammatory, anti-cancer, anti-hypertensive, anti-aging, and immunomodulatory effects (see Table 1), positive impacts have been reported upon the central nervous system, eyes, and brain [31,66]. The astaxanthin market was first dominated (>95%) by synthetic astaxanthin, obtained from petrochemical derivatives. Companies such as DSM in the Netherlands, BASF in Switzerland, and Zhejiang NHU in China then accounted for most supply [67,68]. However, natural astaxanthin derived from *Haematococcus pluvialis* proved superior to its synthetic counterpart [75]. In fact, the intracellular antioxidant capacity of naturally occurring, esterified astaxanthin from *H. pluvialis* exhibits a potency that is 90-fold that of synthetic astaxanthin, besides its being devoid of toxicity. Synthetic astaxanthin appears in free form, while naturally occurring astaxanthin derived from *H. pluvialis* is found either conjugated to proteins (such as in salmon muscle or lobster exoskeleton), or esterified with one or two fatty acids—which stabilize the molecule against oxidation, while facilitating absorption and expression of bioactivity [69,70]. A rising awareness among customers, coupled with increasingly strict regulations pertaining to use of synthetic derivatives has been instrumental to expand market demand for natural astaxanthin. In the case of microalgae, *H. pluvialis* [76] and *Chlorella zofingiensis* [77], together with *Chlorococcum* sp. account for the major natural sources, while only the former has received FDA approval for use as human nutritional supplement, with the others being primarily used as aquaculture feed. The antioxidant properties of astaxanthin are believed to play a key role upon several desirable features, such as protection against ultraviolet-light photooxidation, inflammation, cancer, *Helicobacter pylori*-mediated ulcers, and age-related diseases and aging at large, or else promotion of immune response and liver function, as well as heart, eye, joint, and prostate health [75]. Phycocyanin is another pigment, holding a unique blue color, readily available for food formulation purposes and economically attractive [78]. Although comparative cost evaluation and life cycle assessment of pigments produced auto- and heterotrophically by microalgae have been made available, the lack of trustworthy information specifically on food uses has been outlined [79]. Microalgal cells are also rich in vitamin A, C, E, and K, thiamine, pyridoxin, riboflavin, nicotinic acid, biotin, and tocopherol. Applications are consolidated at the immune system level, encompassing for instance antioxidant activity, cell formation, and blood coagulation [78,80].

The list of health benefits derived from consumption of functional foods containing microalgal ingredients has been steadily increasing. The apparently contradictory nature of some of the results conveyed by the literature may be a consequence of differences in geographical origin, harvesting period, (aqueous) medium characteristics, genetic variability, post-harvest conditions, and method of extraction (including type of solvent used) [81]. Interaction of microalgae with intrinsic or extrinsic properties of foods where they appear—namely pH, fat, protein, water content, and oxygen concentration, and upon preservation thereof, still needs a deeper mechanistic elucidation. Research and development efforts have already solved a number of technical extraction issues, when attempting to adapt microalgal biomass to market specifications of final foods. Alternative forms of such additives include liquid for beverages, powder for flour-based products (e.g., bread, biscuits, pasta), oil for fatty foods (e.g., mayonnaise), and tablets and capsules for food supplements—as tabulated in Table 1.

Sensory aspects play a key role in determining consumer acceptance of foods. Hence, one of the main problems regarding application of microalgal biomass in food products is stability, namely when exposed to more or less severe processing conditions [13]. For instance, powders traditionally obtained from microalgae cultured photosynthetically in outdoor ponds or in indoor photo-bioreactors exhibit too dark a green color (arising from chlorophyll), along with a strong unpleasant taste that may compromise their use in food formulation. Efforts have accordingly been directed toward bleaching of microalgal whole biomass, meant to reduce such an unappealing trait [82]. As shown in Table 1, sensory aspects of functional foods containing microalgae have chiefly pursued higher stability of color, and/or improvement of flavor and texture. This strategy, focused on the additive rather than the basic food matrix itself, circumvents the difficulty in changing food eating habits—especially knowing that Europeans, with long lasting cultural motivations, strongly resist food innovation and diversification [13].

## 3. Microalgae as Functional Feed

Increasing demand for meat by a rising population will become particularly dramatic in the coming decades, because dedicated soybean food crops—the conventional feedstuff for animal feeding, will occupy an increasing fraction of arable land [83]. A few studies have meanwhile unfolded interesting results on use of microalgae as animal feed, in terms of quality and performance of final product (as depicted in Table 2).

Feed enriched with small amounts of microalgal biomass contributes positively to animal physiology, by improving their immune response, disease resistance, and gut function—besides enhancing antiviral and antibacterial protection, as well as increasing reproductive performance, feed conversion and weight gain [84,85]. In particular, use of *Chlorella vulgaris* to feed dairy cattle actually alters the fatty acid profile of milk, by reducing the amount of saturated fatty acid residues and concomitantly increasing the proportion of DHA [86]. Addition of microalgae to feed designed for lambs and horses increased the fatty acid content of the resulting meat [87,88], while inclusion of *Arthrospira platensis* in feed for pigs and poultry improved weight gain [89]—even though at the expense of a lower rate of feed conversion [85]. Genera of microalgae other than *Arthrospira*, *Chlorella*, and *Dunaliella* have been employed by the feed industry. Diets containing microalgae have also been tested in animals besides fish and chicken, with notable degrees of success (see Table 2). The improved quality of the meat obtained, in terms of flavor, color or texture, has favorably contributed to consumer acceptance.

Inclusion of microalgae in aquaculture feeding represents a promising avenue for expansion of the animal production sector, as a sustainable, environment-friendly alternative to classical land agriculture or cattle raising. To date, astaxanthin is mainly utilized in the aquaculture industry as feed additive, to enhance the color of farmed fish and shrimp [90], along with production of good quality seafood for consumption at large [79]. It has been reported that addition of astaxanthin improves growth rate and survival of larvae in aquaculture, as well as reproductive performance and egg quality of aquatic animals owing to its potent antioxidant activity. It has also proven effective toward enhancement of resistance to, and immune response against infectious diseases in farmed fish [91]. From a commercial point of view, application of microalgae as feed in aquaculture is limited by the high costs of production upstream; concentration and storage [17] do indeed still hamper their economic feasibility.

## 4. Microalgae as Probiotics

Emergence of microbiota resistant to conventional drugs and antibiotics has urged alternative strategies to fight them. On the other hand, nano-encapsulated multiplex supplements are still expensive and inconvenient to use. Therefore, simple, low-cost approaches to achieve health benefits in a preventive manner have resorted to supplementation with probiotics [92]. The term probiotic comes from the Greek words “*pro*” and “*bios*”, meaning “for life”—and has classically referred to living microorganisms that naturally help improve health of the host organism at large, when administered in appropriate quantities and as part of a balanced diet (FAO, WHO).

Several studies have unfolded promising results of probiotics, when included in the diet, against various enteric pathogens, due to the unique ability of the former to compete for medium nutrients and adhesion sites, alienate pathogens by secreting antibacterial substances (e.g., bacteriocins, organic acids), and produce antitoxins. When provided at viable numbers above a minimum threshold, probiotics are also able to modulate the autoimmune system, regulate body allergic response, reduce blood pressure, normalize cholesterol and glucose levels, alleviate constipation, and reduce proliferation of cancer cells [118,119].

Probiotic lactic acid bacteria (LAB) were primarily isolated from infant feces, and accordingly considered as an important part of their intestinal microbiota. It was first suggested that they were acquired through oral contamination with maternal LAB during transit through the labor channel. However, recent molecular studies have shown that LAB colonization is not significantly related to the delivery method (i.e., vaginal delivery versus cesarean section). Instead, breast milk was the major source of maternal LAB to the infant gut. As a consequence, probiotics were pioneered as additives by the dairy industry. An increased demand from vegetarian populations, complemented by awareness by consumers of the contribution of milk cholesterol to atherosclerosis and milk protein to allergy have expanded demand for non-dairy probiotic products, as is notably the case of fruits and vegetables. Technological advances have supported changes in structural features of plant matrices, via controlled modification of their components relevant for their eventual roles as food [118]. The resulting taste profiles are indeed appealing to several age groups, and such products are further perceived as healthy foods. Examples include fermented fruits and vegetables [120,121], fruit juices and other beverages [122], and table olives [123,124,125].

The most common probiotics employed in the food industry are bacteria, and yeasts to a lesser extent. They are chiefly derived from human sources and/or animals, and belong mainly to genera *Lactobacillus* and *Bifidobacterium*, seconded by *Streptococcus, Lactococcus,* and *Saccharomyces* [92]. At present, no reliable scientific evidence pertaining to the use of microalgae as probiotics in food is available in the literature.

Centrifugation has been successfully applied to prepare probiotic concentrates, despite its limitations—as said process involves exposure of cells to high centrifugal and shear forces that damage cell structure. Furthermore, processing of large culture volumes is time-consuming and requires expensive equipment, namely specialized continuous centrifuges [126]. Research on post-harvest preservation is still required though, in attempts to extend shelf life beyond 4–8 weeks.

Aquaculture is an important sector aimed at fulfilling nutritional food demand, especially when depletion of natural stocks will lead to upper limits be imposed on sea captures. Meanwhile, disease outbreaks have become a major problem in aquaculture, likely to cause huge economic losses to aquaculture plants. Use of costly chemotherapeutic drugs for treatment holds negative impacts upon the aquatic environment. Hence, a growing impetus exists to find alternatives for control of infectious agents and treatment of diseases, which are safe, non-antibiotic based, and eco-friendly as this is notably the case in the use of probiotics as a food component. The term probiotic in feed farms has been associated to live or dead microorganisms, or components thereof that improve the hosts or the environmental microbial balance when administered via the feed or to the rearing water [127]. An extensive set of research findings have been reported encompassing the effect of bacterial probiotics upon fish health [128].

Several studies claim that the addition of living (probiotic) microalgae to feed will improve health and survival of the marine animal at stake, yet scientific evidence for a probiotic benefit itself is lacking in most of them—as the fate of microalgae in, and their effect upon the microbiota of the gut of said animals has not been ascertained. Remember this is one of the requirements for labeling as a probiotic. For instance, inclusion of *Nannochloropsis oculata* in the diet of seahorse (*Hippocampus reidi*), or *Chaetoceros* sp., *Pavlova* sp., and *Isochrysis* sp. (as such, or combined with each other) in the diet of oyster (*Pinctada margaritifera*) improved survival by reducing the viable count of bacterial pathogens [129,130]. Neyrinck et al. studied the potential hepatoprotective effect of *Spirulina* sp. in aged mice, and its relation to modulation of the gut microbiota [131]. Oral administration of a *Spirulina* sp. was apparently able to modulate the gut microbiota, and to activate the immune system therein—a mechanism eventually leading to improvement of hepatic inflammation. This unfolds the possibility of using microalgae as a new therapeutic tool to preserve a healthy gastrointestinal microbial community, and consequently to enhance their beneficial effect upon immune function, by taking advantage of gut microbe-host interactions [131]. Regunathan and Wesley proved that reduction of pathogenic bacteria in the gut of Indian white shrimp (*Fenneropenaeus indicus*) can be controlled by dietary inclusion of (living cells of) *Tetraselmis suecica* [132]. Nimrat, Boonthai, and Vuthiphandchai examined the effects of probiotic forms (microencapsulated versus freeze-dried cells), probiotic nature (as bacteria, yeasts, or microalgae) and probiotic administration (or supplementation as either water additive or through probiotic-enriched *Artemia*) upon growth, survival rate, and viable microbial numbers of Pacific white shrimp (*Litopenaeus vannamei*) at various stages [133]. Their study suggested that incorporation of microalga *Chaetoceros* sp., in microencapsulated or freeze-dried forms, in water additives, or probiotic-enriched *Artemia* brings about an effective probiotic outcome upon *L. vannamei*—as concluded from enhancement of survival and growth rates, and concomitant increase in numbers of beneficial microbes in the gut of shrimp at larval and post-larval stages, as well as in culture water. This hypothesis was worked out in another three studies focused on the effect of those microalgae on shrimp (*Artemia* sp.) where the reduction achieved in bacterial load was claimed to be of a probiotic nature [134,135,136]. The chance for success may still be enhanced by application of mixed probiotics [137].

Marine hatcheries have easy access to such microalgae as *Tetraselmis* sp., *Chlorella* sp., and *Dunaliella* sp., and handling thereof is straightforward. Therefore, addition of said microalgae to live feed could entail an effective vehicle of otherwise probiotic strains. However, this would require addition of such probiotic strains during the enrichment process, as only very high numbers thereof play a significant role upon the bacterial load in the gut of marine animals. Therefore, this approach is difficult to implement in practice.

Despite the aforementioned pieces of evidence, reliable scientific information that living microalgae do act as true probiotics in aquaculture farms, specifically in controlling pathogenic bacteria, remains scarce at present. In fact, the changes in environmental microbiota and culture conditions make it difficult to prove that live microalgae used as feed will actually control growth of pathogenic bacteria in the gut of the animals fed. Therefore, more detailed investigation is necessary to clarify those points, which may even lead to development of novel probiotic strategies to prevent diseases in aquaculture. Remember that a microorganism can be called a probiotic only after successful completion of the screening stages discussed above [138].

If the probiotic is to be eventually commercialized, analysis of economic feasibility is a must upon successful completion of the in vivo trials. Reliable information of this kind, encompassing different product formulations, packaging options, and dosing recommendations is still in need. In any case, probiotic bacteria-based products are already available in the market. This is the case of a blend of (probiotic) *Bacillus subtilis*, *Bacillus licheniformis,* and *Bacillus pumilus*, traded as *Dans’ Feed with Probiotics* (Aquafauna Bio-Marine, Hawthorne CA, USA).

## 5. Microalgae as Prebiotics

Carbohydrates are the major products derived from photosynthesis and carbon fixation metabolism. However, the chemical profile of, and the metabolic routes involving carbohydrates (mainly starch and cellulose) may differ significantly from species to species of microalga [139,140,141]. In particular, cell wall thickness and composition depend on microalgal species, growth conditions and stage of growth [142].

The cell wall polysaccharides of microalgal biomass can be partially hydrolyzed. This procedure is widely employed in the food and feed industry to produce non-digestible oligosaccharides, which fulfill important roles in human and animal health and nutrition. In fact, microalgal oligosaccharides cannot be fermented at all (or, at least, not completely) by the regular intestinal microbiota of humans [140,143] or animals [144,145]. However, they selectively stimulate growth and activity of specific beneficial bacteria (e.g., *Lactobacilli* and *Bifidobacteria*) if present in the colon, thus contributing to improve the host’s health—in which case, they will act as prebiotics [119,140,143,146]. More specifically, a prebiotic may be seen as an indigestible food compound that effects specific changes in composition and/or activity of gastrointestinal microbiota, thus conferring benefits upon the health of the host as a whole [140].

A few studies suggest that prebiotics exhibit such desirable features as alleviation of irritable bowel syndrome [119], action as antinociceptive agent and peripheral analgesic, and contribution to neovascularization and hepatoprotection [147], besides acting as virucidal, antibacterial, antifungal, anti-inflammatory, immunomodulatory, anticoagulant/antithrombotic, antiproliferative/tumor suppressor, antilipidemic, and hypoglycaemic agent, as well as apoptotic and hypotensive agent [119]. Despite the benefits observed in animal studies, reliable data pertaining to humans are quite scarce. This realization calls for more extensive research on the mechanism of action of prebiotics toward human health and clinical nutrition [146,148]. This is indeed a challenging task, due to the complexity of human gut microbiome, the composition of which can be influenced by a myriad of parameters, ranging from host genetics and state [149] to environment and lifestyle [150].

It has been known for some time that pathogenic and beneficial bacteria coexist in the gastrointestinal tract. Current research trends have dealt chiefly with changes in their balance, namely caused by presence of prebiotics, leading to reduction of (potentially) dangerous bacteria while favoring development of other beneficial ones and accordingly strengthening resistance to infections, or reducing risk of colon cancer and development of obesity. Furthermore, prebiotics have been shown to increase calcium and magnesium absorption, influence glucose levels, and improve plasma lipids [7].

The food industry has always sought more efficient, more sustainable, simpler, and less expensive processes toward application on a large scale. However, production of prebiotic oligosaccharides can be constrained by their structural complexity, so the associated costs may compromise competitiveness of industrial production. On the other hand, prebiotic oligosaccharides can be found in conventional agrofood sources or, alternatively, be produced via enzymatic synthesis from disaccharides or hydrolysis of polysaccharides. Seaweeds and marine microalgae are another relevant (although indirect) source of oligosaccharides, in the form of polysaccharides that will eventually degenerate onto oligosaccharides. They are not broken down by digestive enzymes in the upper part of gastrointestinal tract, while bearing unique biochemical and fermenting features [119]. Conversion of polysaccharides to oligosaccharides may resort to such methods as ultrasound, microwave, free radicals generated by Cu^2+^, Fe^2+^, or H_2_O_2_, hydrolysis by phosphoric acid, and thermal-acidic hydrolysis with dilute HCl [148]. Physical techniques (e.g., ultrasound and microwave) usually exhibit lower, or no side effects at all. In addition, they are not toxic, and quite effective from the points of view of energy and time consumption [140]. Compounds from microalgal sources that possess prebiotic properties include inulin, galacto-oligosaccharides, xylo-oligosaccharides, agarose-derived oligosaccharides, neoagaro-oligosaccharides, alginate-derived oligosaccharides, arabinoxylans, galactans, and β-glucans [151].

Previous studies have shown that *Arthrospira platensis* has a positive effect upon viability of such bacteria as *Lactobacillus casei*, *Streptococcus thermophilus*, *Lactobacillus acidophilus,* and *Bifidobacteria* when part of the intestinal flora [18,143,152,153,154], while such pathogenic bacteria as *Proteus vulgaris*, *Bacillus subtilis* and *Bacillus pumulis* have been suppressed during in vitro studies. When added to yoghurt, *Spirulina* sp. promoted growth of *L. acidophilus* and *Bifidobacteria,* while *Isochrysis galbana*, characterized by high concentrations of both soluble and insoluble fibers, appears promising as a prebiotic. This was concluded from the increase in the number of lactic acid bacteria in the feces of rats, when previously treated with the said microalgae [18,38,52,143].

Specific biological functions played by microalgal species have been related to their sugar complexes, as is the case of *Chlorella pyrenoidosa* and *Chlorella ellipsoidea.* Glucose, and a variety of mannose, galactose, ramnose, N-acetylglucosamine, N-acetylgalactosamine, and arabinose residues are in fact present. Said complexes possess immuno-stimulatory, and even anti-proliferative effects against *Listeria monocytogenes* and *Candida albicans*. Another carbohydrate derived from *Chlorella* sp. that holds immune-stimulatory activity is β-1,3-glucan, namely as free radical scavenger. In addition, it reduces lipid levels in the blood. Polysaccharides extracted from *Porphyridium* sp. and *Nostoc flagelliforme* have also proven effective against Herpes simplex virus [7].

Use of microalgae as prebiotics by the food industry has, however, been restricted so far to dairy products (see Table 3). This is expected since such products are the first vehicles of probiotic (bacterial) strains, as explained above. As also outlined before, vegan consumers are looking for non-dairy matrices, and technological advances have already permitted tailor-made modification of food components in fermented fruits and vegetables. Oligosaccharides derived from plant material have been claimed to hold a prebiotic potential [155,156,157] and some are already available in commercial form (e.g., Prebiotin^®^). Therefore, there is a window of opportunity for development of prebiotics from microalgae, to be eventually applied to lactic acid-fermented foods other than yogurt or cheese.

In aquaculture, immunostimulants have been exploited toward strengthening of the immune system of fish subjected to stress. Those compounds promote, in general, cell activity, and proliferation of such leukocytes as monocyte-macrophages and neutrophils, as well as phagocytic activity and secretion of immune mediators (e.g., cytokines). Paramylon (a linear β-1,3 polymer of glucose) is one such immunostimulant used in aquaculture, as depicted in Table 4. These prebiotic compounds are incorporated as supplements in the feed (but not together with live probiotics therein). Representative examples encompass mussels [158], Atlantic salmon [159], matrinxã [160], and red drum [161]. A parallel application entails addition of β-glucan derived from *Euglena* sp. to feed for poultry, cows, horses, dogs, cats, reptiles and birds, as well as valuable exotic animals kept at zoos or in aquariums [162].

Beta-glucan derived from yeast (especially *Saccharomyces cerevisiae*) has to date been the most successful prebiotic in the market. It is commercially available as WellMune^TM^ by Biothera Corporation (Eagan, MN, USA), BetaGlucans by BioTec Pharmacon (Tromsø, Norway), and as Macrogard^TM^ by Immunocorp (Werkendam, Neteherlands). As an exception to this rule is paramylon (available commercially as Algamune™ by Algal Scientific Corporation (Plymouth, MI, USA)) [161], which has been produced to high purity levels departing from preparations of β-glucans from yeasts, curdlan from Gram-negative bacteria, laminarin from brown seaweeds, or scleroglucan from fungi [162]. Despite the great probability of finding prebiotic compounds among marine-derived saccharides, the exact composition and substituent distribution needed for effective activity remains largely to be explored. Hence, deeper investigation of these compounds will likely provide novel insights on the specific structures required to enhance prebiotic activity.

## 6. From Research Findings to Actual Commercialization

Production at the industrial scale of microalgae-based products with a functional role emerges as an opportunity, given the information already made available by research studies. This may support gains of market share in the bioactive molecule segment, dominated so far by synthetic molecules, or molecules extracted from animal and plant sources (see Table 5).

In the early 1960s, single cell protein (SCP), or protein-rich biomass, for feed was the main product targeted by industry, while applications directed toward food and prophylactic uses appeared only later. Production of microalgal pigments boomed in the 1980s, through cultivation of *Dunaliella* sp. and *Haematococcus* sp. with a focus on β-carotene and astaxanthin, to be used as additives in food or feed. Production of PUFAs, chiefly docosahexaenoic acid (DHA) and eicosapentaenoic acid (EPA), started in the early 1990s – for eventual use in aquaculture feed and enrichment of nutritional products. Despite some microalga-based products bearing a relatively long tradition, large-scale commercial production is still in its infancy. Commercial facilities for microalga production are scattered worldwide (see Table 5) while commercialization is dominated by North America and Asia, with rather poor inputs by Europe, North Africa, and Oceania.

Since most microalga-based products at present are intracellular, biomass production rate and yield represent the main criterion to ascertain techno-economic feasibility. Bulk production of proteins, carbohydrates, and lipids via microalgae as cell factories is not foreseen in the short run, owing to the high production volumes expected. An economy of scale therefore plays a crucial role in capital and operational expenditures associated to the processes, with substantially high fixed capital expenditures and labor costs [174]. Additionally, high-value products from microalgae are normally intended for human or animal consumption, so their manufacture processes have to abide to a range of regulations and standards. Labelling issues worsen the problem, as they vary widely from country to country, and can add to the overall cost of manufacturing a saleable product [11,174]. Recent studies have, however, taken advantage of biorefinery strategies and treatment of wastewater to relieve net processing costs, while enhancing sustainability of microalga production.

## 7. Sustainability Issues

A number of studies have recently addressed the market success of microalgae in the food/feed industries based on sustainability considerations [198,199,200,201,202]. One interesting way to alleviate the high costs of producing microalgal biomass for animal feed is to first extract their lipids for production of biodiesel, due to their intrinsically high lipid contents, and then process the remainder lipid-free material to obtain protein-rich products [203]. Such a possibility may extend to the situations where biomass is obtained from wastewater as growth medium in the first place, as long as it is clearly demonstrated that the final biomass is free of pathogens, toxins, and harmful residues of any kind, and thus safe for feed uses. For instance, Trivedi et al. [201] showed that *Chlorella vulgaris* can effectively be cultivated in untreated wastewater from fish processing industry, without the need to add extra nutrients, and used downstream for biodiesel production, before ending up as protein-rich feed. *Botryococcus*, *Chlorella,* and *Scenedesmus* appears as the most promising genera in this endeavor [202], while extraction should resort, as much as possible, to green solvents. Scaleup of microalga cultivation in wastewater apparently does not hamper biomass productivity [198] and circumvents the need for an intermediate step of physical extraction, or plant-mediated concentration of the said nutrients prior to incorporation in the final feed. Alternative uses under scrutiny include fermentation of leftover biomass for production of bioethanol or biomethane. Enhancement of lutein productivity to be extracted prior to feed use of the left biomass, can also resort to flue gas as a base nutrient. Quantitative removal of nitrate, nitrite, ammonium and carbon dioxide, dissolved in the aqueous medium upon bubbling, was reported elsewhere [200].

In more quantitative terms, microalgae-based processes can reduce the energy consumption of conventional wastewater treatments by about one half, while allowing recovery of up to 90% of the nutrients present therein. When production of microalgae is coupled with nutrient recovery from wastewater, the associated production costs will lie below 1 €/kg—the threshold normally accepted for economic feasibility [199]. However, there is still room for improvement of the technology currently available, in terms of photobioreactor-based production and harvesting procedures, in attempts to reduce land requirement and hydraulic retention time [198]. Said technology also has to be adapted to a wide range of wastewater sources, from domestic sewage to cattle manure. Target figures indicate a need of 450 ton of C, 25 ton of N, and 2.5 ton of P per ha and per year to support a biomass productivity of 200 ton ton/ha·yr, while current levels are still well below and improvements will likely need a higher efficiency in light utilization. At the same time, sequestering of atmospheric CO_2_ to serve as source of C constitutes a major environmental advantage of phototrophic microalgae, which will strengthen their competitive advantage, especially as environmental legislation becomes stricter and stricter. Another advantage of resorting to wastewater, as a raw material, comes from the high-water consumption nature of microalga-mediated processes, besides energy demands and requirement for bioavailable forms of nitrogen [202]. One well-documented example is production of high-value lipids, namely essential fatty acids with claimed health benefits (e.g., PUFAs-ω3). Fernandez et al. [198] claimed that coupling of biomass production with nutrient recovery from waste will constitute an irreversible trend in the coming years, in a continuing quest for economic feasibility and to further search for better performing strains and consortia thereof.

## 8. Conclusions

Scientifically validated evidence that compounds in microalgal viable or unviable biomass play functional roles when used as, or incorporated in food or feed has been recently increasing. This mini review attempted to provide a useful, yet concise update on scientific application and commercialization in the field. Biotechnological research in this area is promising, and likely to offer new types of biologically active compounds that are relevant in attempts to decrease frequency of occurrence and severity of chronic diseases, in both humans and animals. The fast growing body of scientific and technological information unfolds indeed a great potential for large-scale production in the near future.

Nutritionally speaking, microalgal proteins are comparable to plant proteins, yet their commercial development has been hampered by higher production costs, technical difficulties in extraction and refining, and sensory and palatability issues when attempting to formulate novel food products.

Further to the technological and economic challenges posed to production of microalgae at large scale for food and feed uses, the sustainability issue is to be appropriately addressed. Taking advantage of microalga cell factories to also produce biodiesel, or use of wastewater as broth for microalga growth are but two possibilities likely to prove economically feasible.

Studies on uses of microalgae as probiotics in food are quite scarce, unlike what happens with probiotic bacteria and (more recently) yeasts. Based on the possible use of microalgae as probiotics in the feed industry, resources are to be focused on how to promote survival and growth of microalgae, and how to deliver such alive beings using food as a vehicle. Reduced application of microalgae as prebiotics in the food industry is also apparent, despite their potentially beneficial effects upon human health, via control of host’s gut microflora, thus further research efforts are accordingly needed.

Considering the above facts, microalga-based products still face technological and economic difficulties to win the market battle. However, this situation is likely to be reversed due to their potential features, as conveyed by a fast growing body of scientific information generated by academy, startups, and multinational companies. In any case, quality assurance, safety, regulatory and labeling issues, and environmental impact have to be duly addressed when attempting to bring a microalgal product to the market which will necessarily extend the time period prior to commercial production.

## Figures and Tables

**Table 1 marinedrugs-17-00312-t001:** Potential industrial applications of microalgae in functional food, sorted by type of food, commercial form of biomass, and bioactive compound.

Microalgae	Food		Commercial Form of Biomass	Bioactive Compound	Health Benefit	Reference
Genus/Species	Product	Sensory Effect				
*Chlorella* sp. *Sprirulina* sp.	Milk	Improved flavor and mouthfeel	Powder or liquid	Protein, PUFA-ω3, EPA *, DHA **	Reduced risk of anemia	[32]
*Arthrospira platensis*	Yoghurt	Improved texture and viscosity	Extract	Phycocyanin	Anticancer; antioxidant and anti-inflammatory	[33]
*Arthrospira platensis Chlorella* sp.	Cheese	Improved texture	Powder	Protein, carbohydrates, PUFA-ω3	Anticancer; reduced risk of gastric ulcers, constipation, anemia, hypertension, diabetes, infant malnutrition, neurosis	[34,35]
*Spirulina* sp.	Alcohol-free beverage	Improved color and sour taste	Powder or liquid	Protein, chlorophylls, phycocyanin	Improved immune and lymphatic systems, protection against cancers and ulcers	[36,37]
*Arthrospira maxima* *Chlorella protothecoides* *Haematococcus pluvialis*	Desserts	Improved color and stability	Powder or flour	Protein, vitamins, minerals	Antioxidant activity, prevention of constipation	[38,39]
*Arthrospira platensis Chlorella vulgaris* *Hematococcus pluvialis Phaeodactylum tricornutum* *Tetraselmis suecica*	Cookies and biscuits	Improved color, stability and texture	Powder or flour	Protein, PUFA-ω3, EPA, DHA, astaxanthin	Antioxidant activity	[12,40,41,42,43,44]
*Arthospira platensis Chlorella* sp.	Bread and cookies	Improved flavor, texture and appearance	Powder or flour	Protein, vitamins, minerals	Reduced fat and cholesterol levels, induced satiety	[45,46,47]
*Dunaliella* sp. *Spirulina* sp.	Miso	Slightly seaweed taste	Powder	Protein, vitamins, minerals	Antioxidant activity	[48]
*Chlorella* sp. *Sprirulina* sp.	Koji	No flavor or smell	Powder	n.a. *****	Improved immunity and blood pressure	[49]
*Dunaliella salina*	Pasta	Improved color and texture	Powder	Protein, carotenoids	Antioxidant activity	[50]
*Diacronena volkianum Isochrysis galbana*	Pasta	Improved color, flavor, texture and firmness	Powder	Protein, PUFA-ω3, EPA, DHA, carotenoids	Protection against gastric ulcers, prevention of constipation, reduced anemia and diabetes, improved blood pressure	[51,52]
*Arthrospira* *maxima* *Diacronena volkianum* *Haematococcus pluvialis*	Vegetarian food gels	Improved color and firmness	Gels	PUFA-ω3, EPA, DHA, GLA ***, carotenoids	Antioxidant activity	[53]
*Chlorella vulgaris* *Haematococus pluvialis*	Emulsions or vegetarian mayonnaise	Improved color and stability	Oil or emulsions	Protein, carotenoids	Antioxidant activity	[4,54]
*Chlorella vulgaris*	Soybean oil	Improved color and stability	Oil	Carotenoids	Antioxidant activity	[55]
*Arthrospira platensis*	n.a.	n.a.	Oil	Carotenoids	Antimicrobial and antiviral activities	[56]
*Dunaliella salina*	Culinary condiment with sea salt	Improved flavor	Powder	Carotenoids	Antioxidant activity	[57]
*Chlorella* sp. *Schizochytrium* sp. *Thraustochytrium* sp.	Food supplement	n.a.	Powder, flour, tablet or liquid	Proteins, PUFA-ω3	Prevention of constipation, induction of satiety	[58,59]
*Dunaliella* sp. *Phaeodactylum tricornutum* *Nannochloris* sp. *Nannochloropsis* sp.	Food supplement	n.a.	Capsules	Protein	n.a.	[60]
*Haematococcus pluvialis*	Food supplement	n.a.	Capsules	Astaxanthin	Improved eye and brain health, UV protection and skin health, anti-coagulatory and anti-inflammatory effects in diabetes, immune system modulation, cardiovascular health	[61,62,63]
*Parietochoris incisa*	Food supplement	n.a.	Powder or tablet	ARA ****	n.a.	[64]
*Tetraselmis suecica*	Food supplement	n.a.	Extract	n.a.	Prevention of obesity and diabetes	[65]

* EPA—Eicosapentaenoic acid, ** DHA—Docosahexaenoic acid, *** GLA—Gamma-linolenic acid, **** ARA—Arachidonic acid, ***** n.a.—Information not available.

**Table 2 marinedrugs-17-00312-t002:** Potential industrial applications of microalgae in functional feed, sorted by type of animal, resulting food product, and bioactive compound.

Microalgae	Feed	Resulting Food		Commercial Form of Biomass	Bioactive Compound	Health Benefit	Reference
Genus/Species	Animal	Product	Sensory Effect				
*Schizochytrium* sp.	Cow	Meat	n.a ***	Powder	PUFA-ω3, EPA *, DHA **	Improved cardiovascular, brain and eye systems	[93]
*Chlorella vulgarisa* *Spirulina* sp.	Piglet	Meat	n.a.	Powder or spray	Cu	Increased nutritional properties	[94]
*Arthrospira platensis Isochrysis* sp.	Lamb	Meat	Improved color, (not so intense) odor and flavor	Powder	Protein, PUFA-ω3	Prevention of cardiovascular diseases	[95,96,97,98]
*Arthrospira platensis Schizochytrium* sp.	Rabbit	Meat	n.a.	Powder	PUFA-ω3, γ-linolenic acid	Anti-inflammatory activity, increased nutritional properties	[99,100]
*Arthrospira platensis Chlorella vulgaris* *Staurosira* sp. *Schizochytrium* sp.	Chicken	Meat	Improved color (yellowness of flesh, and redness of liver)	Powder or spray	PUFA-ω3, EPA, DHA	Antibiotic activity, reduced risk of chronic diseases, improved well-being	[101,102,103,104,105,106]
*Chlorella vulgaris*	Pekin duck	Meat	Improved color (yellowness of flesh)	Fermented	Protein	Improved immunity	[107]
*Arthrospira platensis Nannochloropsis gaditana*	Hen	Egg	Improved color (yellow to orange)	Powder or spray	PUFA-ω3, EPA, DHA, carotenoids	Prevention of cardiovascular diseases, anti-inflammatory, antihypertensive, anticancer, antioxidant, antidepressing and antiaging activities	[108,109,110]
*Porphyridium* sp.	White Leghorn chicken	Egg	Improved color (yellow to orange)	Freeze dried	PUFA-ω3, EPA, DHA, γ-linolenic acid	Improved nutritional properties	[111]
*Dunaliella* sp.	Shrimp	Meat	n.a.	Freeze dried	Carotenoids	Antioxidant activity, improved immunity	[112]
*Tetraselmis chuii*	Shrimp	Meat	n.a.	Freeze dried	Astaxanthin	Antioxidant activity	[113]
*Nanofrustulum* sp. *Tetraselmis* sp.	Atlantic salmon	Meat	n.a.	Powder	Protein, lipids	Improved nutritional properties	[114]
*Haematococcus pluvialis*	Salmon and trout	Meat	Improved color	Powder	Astaxanthin	Antioxidant activity	[115]
*Arthorospira platensis*	Coral trout	Meat	n.a.	Pellet	Protein, lipids	Improved nutritional properties and immunity	[116]
*Arthrospria maxima Chlorella vulgaris* *Haematococcus pluvialis*	Koi carp goldfish	Food supplement	Improved color (red hue)	Powder	Carotenoids	Antioxidant activity	[117]

* EPA—Eicosapentaenoic acid, ** DHA—Docosahexaenoic acid, *** n.a.—Information not available.

**Table 3 marinedrugs-17-00312-t003:** Potential industrial applications of microalgae as prebiotics in food, sorted by type of food.

Microalgae			Target Probiotic Bacteria	Food		Health Benefit	Reference
Genus/Species	Commercial Form of Biomass	Bioactive Compound		Product	Sensory Effect		
*Arthrospira platensis Chlorella vulgaris*	Powder	Glucose, rhamnose, mannose, xylose and galactose	*Lactobacillus acidophilus* *Bifidobacterium lactis* *Lactobacillus delbrueckii* *Streptococcus thermophilus*	Yoghurt	Improved color, stability and texture	Prevention of constipation, improved immune system, enhanced absorption of minerals and lactose, reduced cholesterol	[155,163,164,165]
*Arthrospira platensis*	Powder	PUFA-ω6	*Streptococcus thermophilus* *Lactobacillus delbrueckii* spp. *bulgaricus Lactobacillus lactis* ssp. *lactis Lactobacillus acidophilus*	Milk	n.a. *	Improved nutritional properties	[166]
*Cryptheiconidium cohnii*	Powder or freeze dried	Phycocyanin, vitamin C, Se, Zn, Fe, Mg	*Streptococcus salivarius* *Thermophilus* sp. *Lactobacillus delbrueckii* *Lactobacillus bulgaricus* *Bifidobacterium bifidum* *Lactobacillus acidophilus*	Milk	n.a.	Anticarcinogenic and anti-inflammatory activities, improved blood and cholesterol levels	[167]
*Chlorella* sp. *Scenedesmus* sp. *Spirulina* sp.	Powder	Carotenoids, γ-linolenic acid	*Lactobacillus plantarum* *Bifidobacteria*	Cheese	Improved color (green-blue) and texture	Improved nutritional properties	[168]

* n.a.—Information not available.

**Table 4 marinedrugs-17-00312-t004:** Potential industrial applications of microalgae as prebiotics in feed, sorted by commercial form of biomass and animal type.

Microalgae			Probiotic Bacteria	Animal	Health Benefit	Reference
Genus/Species	Commercial Form of Biomass	Bioactive Compound				
*Navicula* sp.	Freeze dried	Oligosaccharides	*Lactobacillus sakei*	Pacific red snapper *(Lutjanus peru)*	Improved immune system and antioxidant activity	[169]
*Phaeodactylum tricornutum Tetraselmis chuii*	Freeze dried	Protein	*Bacillus subtilis*	Gilthead seabream (*Sparus aurata*)	Improved immune system and increased intestinal absorption ability	[144]
*Arthrospira platensis*	Freeze dried	C-phycocyanin	*Vibrio alginolyticus*	Shrimp (*Litopenaeus vannamei*)	Improved immune system (>lysozyme) and disease resistance	[170]
*Dunaliella tertiolecta*	Freeze dried	β-carotene	*Bacillus* sp.	Shrimp *(Artemia franciscana)*	Improved immune system and disease resistance	[136]
*Arthrospira platensis*	Powder	Phycobilins, phycocyanin, allophycocyanin, xanthophylls and carotenoid	*Pseudomonas fluorescens*	Nile tilapia (*Oreochromis niloticus* L.)	Improved immune system and antioxidant activity	[171]
*Euglena gracilis*	Powder	Paramylon	*Streptococcus iniae*	Red drum (*Sciaenops ocellatus* L.)	Immunostimulant activity	[161]
*Arthrospira platensis*	Powder	Oligosaccharides	*Bacillus subtilis*	Prawn (*Penaeus merguiensis*)	Improved immune system and disease resistance	[172]
*Euglena gracilis*	Powder	β-glucan	*Bacillus licheniformis* or *Bacillus subtilis*	Poultry, cows, horses, dogs, cats, reptiles, birds	Improved well-being and immune system	[162,173]

**Table 5 marinedrugs-17-00312-t005:** Examples of commercial applications of microalgae in food and feed, sorted by microalga.

Microalgae	Food			Feed		Reference
Genus/Species	Main Product	Main Product Application	Industries	Main Product	Industries	
*Arthrospira plantensis*	Phycocyanin	Food colorant and supplement	A4F-Algae 4 Future (Portugal) Blue Biotech (Germany) DIC Lifetec (Japan) E.I.D Parry (India) Necton (Portugal) Ocean Nutrition (Canada)	Feed supplement	Blue Biotech (Germany) Ocean Nutrition (Canada)	[175,176,177,178,179,180]
*Chlorella vulgaris*	Lutein	Food supplement	A4F-Algae 4 Future (Portugal) Algomed (Germany) Buggypower (Portugal) E.I.D Parry (India) Phycom (Netherlands) Chlorella Co. (Taiwan)	Feed supplement	Blue Biotech (Germany) Necton (Portugal)	[175,176,178,179,181,182,183,184]
*Dunaliella salina*	β-carotene	Food colorant and supplement	BASF (Germany) Nikken Sohonsa Co. (Japan) Wonder Care Pvt. Ltd. (India) Solazyme, Inc. (San Francisco)	Feed supplement	Blue Biotech (Germany) Necton (Portugal) Algalimento SL (Canary Islands)	[176,179,185,186,187,188,189]
*Haematococcus pluvialis*	Astaxanthin	Food supplement	AlgaTech (Israel) AstraReal Co. (Japan) Blue Biotech (Germany) Fuji Chemicals (Japan) E.I.D Parry (India) Solix Inc. (USA)	Feed supplement (pigment enhancer for fish)	Blue Biotech (Germany)	[176,178,190,191,192,193]
*Labosphaera incisa*	ARA ***	Food supplement	A4F-Algae 4 Future (Portugal)	n.a. ****	n.a.	[175]
*Nannochloropsis* sp.	EPA and DHA (ω-3)	Food supplement	AstraReal Co. (Japan) AlgaTech (Israel) Cyanotech (US, Hawaii) E.I.D Parry (India)	Feed additive	Blue Biotech (Germany) Innovative Aqua (Canada)	[176,178,189,190,191,194]
*Euglena gracilis*	Paramylon/Linear beta-1,3-glucan	Food supplement	Algaeon Inc. (USA) Kemin Industries (USA) Valensa International (USA)	n.a.	n.a.	[195,196,197]
Phaeodactylum tricornutum	EPA (ω-3), Fucoxanthin	Food supplement	A4F-Algae 4 Future (Portugal) AlgaTech (Israel)	n.a.	n.a.	[175,190]

* EPA—Eicosapentaenoic acid, ** DHA—Docosahexaenoic acid, *** ARA—Arachidonic acid, **** n.a.—Information not available.

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
