# Peer review of "Potential Industrial Applications and Commercialization of Microalgae in the Functional Food and Feed Industries: A Short Review"

_marinedrugs, 2019, doi:10.3390/md17060312_

Reviewer 1 Report

I have put my suggesstions in comments in the pdf file

My main concern regards the need to address in the ms some of the most recent issues related to the process sustainability, the microalgae production for food/feed sustainability...

e.g. feed production in combination with wastewater treatment

the biorefinery approach

innovations in downstream processes

Author Response

MARINE DRUGS-502332

Reply to reviewer 1

POINT 1. I have put my suggestions in comments in the pdf file.

We made the changes in the manuscript document, and marked them with Track Changes.

We responded to the queries on the pdf document, as well as in this document as follows.

Line 14: omit Such. Omitted.

Line 14: Why fiber are bioactive?

This information is found in several articles. One example is:

Papandreou, D.; Noor, Z.T.; Rashed, M. The Role of Soluble, Insoluble Fibers and Their Bioactive Compounds in Cancer: A Mini Review. Food Nutr. Sci. 2015, 06, 1–11.

Abstract The cancer incidence has risen dramatically over the last decades. About 8 million people died globally according to latest reports, which represented almost 40% more than it was 20 years ago. Risk factors for the development of cancer have been found to include smoking, alcohol, drugs, obesity and diet. Fiber intake has shown to exhibit chemoprotective effects on cancer proliferation and metastasis that may seem to be very promising. This article will review the role of different types of fiber such as, cellulose, lignin, pectin and inulin in development and prevention of different types of cancers. This article would also discuss the effectiveness of both types of fiber in cancer prevention.

Line 34 change into 'Microalgae are ancestral organisms (omit living) that constitute the basis aquatic food chains'. Changed as suggested.

Lines 35-38: As the term microalgae refers to eukaryotic organisms this paragraph is not correct and to my opinion should be deleted or changed as follows: microalgae are a phylogenetically diverse group encompassing a number of different phyla and classes of organisms, in some cases cyanobacteria are also included. Changed as suggested.

Line 39: environments. In order to not to repeat the environment terms twice in the same sentence, this was changed to “Microalgae grow well not only in freshwater, seawater and hypersaline environments, but also in moist solids and rocks”.

Line 45: italics. Done.

Line 49: omit own and add biomass. Done.

Line 53: fibers? The term fiber was removed.

Line 53 Minerals, say what? In reference 13, the following minerals are listed: Na, K, Ca, Mg, Fe, Zn and trace minerals, so the text was rephrased as “and minerals (e.g. Na, K, Ca, Mg, Fe and Zn)”

Line 55: what is meant with traditional in this case? do you mean crop plant??? Yes, so the text was changed as suggested.

Line 71: Change into “some extent”. Changed as suggested.

Lines 78-79: Omit “via” and “excessive amounts of”. Done.

Line 79: Change into: freshwater (1word) consumption. Done.

Line 86: Proteins. Done.

Line 86: add “former Spirulina”. Done.

Line 90: italics. Done.

Line 94: Human? Yes, changed accordingly.

Line 97: the. Changed as suggested.

Line 108: add reference. Done.

Line 109: for. Done.

Line 112: algae can also present chl b and variants of chl c. Correct into chlorophylls. Done.

Line 117-118: in standard or stress culture conditions? Stress conditions, and text was modified accordingly. would add 'can' before 'account...Done. in terms of percentual relative to biomass dry weight? percentual relative to biomass total weight, so this was also made clear in the text.

Line 118: Studies also focused on cell accumulation and product yield optimization. Information added, as suggested.

Line 118: of these two green algal species. Information added, as requested.

Line 123: change into levels. Done.

Line 127: changed to “presentation form”

Line 145: italics Done.

Line 159: omits italics. Done.

Line 161: their to its. Done.

Line 162: italics. Done.

Line 205: omit. Done.

Line 205: change to “outdoor”. Done.

Line 205: change interior to indoor. Done.

Line 210: Microalgal biomass. Yes we do. The word “whole” was added to the text.

Line 224: Fatty. Done.

Line 263: The. Done.

Line 305: living. Done.

Line 320: living cells of. Done.

Line 339: living. Done.

Line 335: This statement cannot be generalized for algae the carbon storage product can be different from starch and not localized in the plastids; the cell walls can contain sugars as galactans, urinic acids and not only cellulose,  please avoid genralization and omit as not complete in its present form. We agree, and thus omitted as suggested.

Line 359: Based on your comments “the different composition of cell walls in phylogenetically distant algae is a matter of fact and also a well known topic for phycologists

I would avoid the use of the verb 'appear'

I cannot understand the bit 'but the mechanistic justification of these claims is still lacking'  if I interpret it correctly the reason for this highly different and taxon specific cell wall composition relies of the phylogeny and ecology of the alga/strain used, and obviously on the culture conditions applied we accordingly modified the sentence to “In particular, cell wall thickness and composition depend on microalgal species, growth conditions and stage of growth”.

Line 361: change biomass into cells. Done.

Line 392, 406, 414, 417, 418, 419: italics.

Line 420: Nostac to Nostoc; flumelliform to flagelliforme. In reference 7 the name of Nostac flegelliforme wich was wrong. We wrote the correct name as you suggested.

Line 461: this is a repetition. We agree; it was already in page 79, so we removed it.

Line 462: single cell protein??? What is meant with this? Single cell protein (SCP), or protein-rich biomass.

Line 474: italics

Line 481: We add a topic  called 7. Sustainability issues. Conclusions were changes to topic 8.

POINT 2. My main concern regards the need to address in the ms some of the most recent issues related to the process sustainability, the microalgae production for food/feed sustainability...

e.g. feed production in combination with wastewater treatment

the biorefinery approach

innovations in downstream processes

In view of the main focus elected for this minireview, such side subjects as sustainability of production - including namely wastewater treatment, were referred to a dedicated, new (necessarily brief) section.

Reviewer 2 Report

I have read carefully the manuscript marinedrugs-502332-2 entitled “Potential industrial applications and commercialization of microalgae in the functional food and feed industries: a short review”. The authors of this work present here a mini review they wrote about the potential of microalgae as being used as ingredients for functional food and feed.

My principal remark will concern the main substance of this article that is clearly very interesting, very well documented and constitutes a very serious review work. Only minor remarks should be performed, as english editing. I would recommend therefore this manuscript for publication in the journal marine drugs, into a minor revised form.

Sincerely yours.

Author Response

English language and style are fine/minor spell check required. Done.